# UV-Cured Chitosan-Based Hydrogels Strengthened by Tannic Acid for the Removal of Copper Ions from Water

**DOI:** 10.3390/polym14214645

**Published:** 2022-11-01

**Authors:** Rossella Sesia, Sara Ferraris, Marco Sangermano, Silvia Spriano

**Affiliations:** Politecnico di Torino, Dipartimento di Scienza Applicata e Tecnologia, Corso Duca Degli Abruzzi 24, 10129 Torino, Italy

**Keywords:** hydrogels, chitosan, tannic acid, UV-curing, copper ions adsorption, reusability

## Abstract

In this work, a new environmentally friendly material for the removal of heavy metal ions was developed to enhance the adsorption efficiency of photocurable chitosan-based hydrogels (CHg). The acknowledged affinity of tannic acid (TA) to metal ions was investigated to improve the properties of hydrogels obtained from natural and renewable sources (CHg-TA). The hydrogel preparation was performed via a simple two-step method consisting of the photocrosslinking of methacrylated chitosan and its subsequent swelling in the TA solution. The samples were characterized using ATR-FTIR, SEM, and Folin–Ciocalteu (F&C) assay. Moreover, the mechanical properties and the ζ potential of CHg and CHg-TA were tested. The copper ion was selected as a pollutant model. The adsorption capacity (Q_e_) of CHg and CHg-TA was assessed as a function of pH. Under acidic conditions, CHg-TA shows a higher Q_e_ than CHg through the coordination of copper ions by TA. At an alkaline pH, the phenols convert into a quinone form, decreasing the Q_e_ of CHg-TA, and the performance of CHg was found to be improved. A partial TA release can occur in the copper solution due to its high hydrophilicity and strong acidic pH conditions. Additionally, the reusability of hydrogels was assessed, and the high number of recycling cycles of CHg-TA was related to its high mechanical performance (compression tests). These findings suggest CHg-TA as a promising green candidate for heavy metal ion removal from acidic wastewater.

## 1. Introduction

Recently, close attention has to be paid to wastewater remediation, as contaminants can lead to serious problems for the ecosystem. Among the pollutants of greatest concern are heavy metal ions. They can derive from several anthropogenic sources due to increasing industrialization and urbanization [1]. Heavy metal ion contamination is usually caused by various effluents of many industries, including metallurgy, battery and electric device manufacturing, metal plating, and the use of agricultural products [2]. Copper is used extensively in the electrical industry, as well as in the production of fungicides and anti-fouling paints [3]. Although copper is considered a micronutrient for human beings, an excessive dosage can cause acute toxicity, with sometimes lethal effects [4]. Therefore, the decontamination of heavy metal ions from water is an urgent and mandatory environmental need, since they are non-biodegradable and highly toxic substances that can enter the food chain with bioaccumulative effects [5].

Several techniques for the treatment of contaminated water have been used, including ion exchange [6], chemical precipitation [7], the advanced oxidation process [8], membrane filtration [9], coagulation–flocculation [10], and electrochemical techniques [11]. However, these conventional methods are often expensive or inefficient, and are mainly used for removing the pollutants from diluted solutions. Among the several decontamination strategies, adsorption is the most efficient method for heavy metal-polluted water remediation [12,13]. It is also cost-effective.

The adsorption process is a surface phenomenon, and it is used as a wastewater decontamination method at a post-treatment level (polishing) after pre-filtration to enhance the adsorption efficiency [14]. The process can be categorized as physisorption or chemisorption, depending on the type of interaction between the adsorbate species and the adsorbent surface. Furthermore, the ease of performance and the possibility of multiple reuses, with no by-products, make the adsorption process an excellent choice for wastewater treatment [5,15]. In general, many materials, such as zeolite, clay minerals, activated carbon, silica, and magnetic particles, have been studied as adsorbents to remove metal ions from the aqueous phases [16,17,18,19,20]. In the last few years, a growing research interest has been focused on low-cost adsorbents as alternatives to manufactured adsorbents. Within this framework, the development of polymeric hydrogels, according to the guidelines of green chemistry principles, has shown promising potential to selectively uptake metal pollutants. Hydrogels are three-dimensional, flexible, and porous networks with high swelling properties in water due to the presence of hydrophilic groups such as hydroxyl, carboxyl, and amide [21,22].

Over recent decades, renewable and natural resources have been recognized as promising candidates for water remediation, reducing the environmental impact of the adsorption treatment. For this reason, polysaccharides (e.g., cellulose, starch, hyaluronic acid, alginate, chitosan) have been proposed in environmental applications for heavy metal ions removal due to their distinctive properties, such as biodegradability, biocompatibility, non-toxicity, and low price [22,23,24,25]. The biosorption mechanism is based on physicochemical interactions between hydrogel and metal ions, including electrostatic interaction, metal ion complexation, hydrogen bonding, ion exchange, and coordination/chelation [26]. The experimental conditions (e.g., pH, temperature), the type, and concentration of the pollutant play a significant role in the adsorption efficiency [27].

Among biobased hydrogels, chitosan has been found to be efficient for the uptake of several heavy metal ions, including mercury, chromium, copper, lead [28,29,30], and arsenic [31]. Chitosan is a copolymer of (1,4-β-)-2-acetylamino-2-deoxy-D-glucopyranose and (1,4-β-)-2-amino-2-deoxy-D-glucopyranose (Figure 1a). It is a derivative from the N-deacetylation of chitin, which is the second-most abundant polysaccharide in nature, obtained from crustaceans, algae, fungal biomass, insects, and mollusks [12,32,33].

However, those natural polymer networks exhibit low mechanical strength and poor chemical stability, particularly in acidic media, and thus, are easily destroyed during the decontamination treatment, causing a low adsorption rate [12,22]. To overcome these drawbacks, chemical or physical crosslinking methods have been applied [34]. Nevertheless, the most common chemical networking agents, such as formaldehyde [35], epichlorohydrin [36], and N,N’-methylenebisacrylamide [37], are harmful to human and environmental health, and the physical networking methods using the introduction of ionic interactions, hydrogen bonding, and molecular entanglements are not significantly effective for heavy metal ion removal. In the literature, very few works have studied the use of the photocrosslinking process to produce chitosan hydrogels, an environmentally friendly technique to achieve crosslinked hydrogels through the chemical addition of a crosslinker [38].

In addition, another potential source of inexpensive bioadsorbents are tannins. They are one of the most ubiquitous compounds extracted from terrestrial biomasses and the major source of polyphenolic components. Two classes of tannins are identified: hydrolysable tannins and condensed tannins. Hydrolysable tannins are subdivided into ellagitannins and gallotannins, depending on the type of acid produced during hydrolysis, and tannic acid (Figure 1b) is the most prominent example. The condensed tannins include oligomers and polymers [39,40].

Both types of tannins show a marked affinity towards metals, thanks to multiple adjacent phenolic hydroxyls, and this feature is widely exploited in the field of wastewater treatment, as well as in the recovery of precious metals [41,42,43,44]. However, since tannins are water-soluble compounds, they require a chemical immobilization process, which is usually expensive and toxic due to the use of crosslinking agents such as formaldehyde, hexamethylenetetramine, glyoxal, and glutaraldehyde [45].

So far, however, there has been little discussion about the capacities of tannins and biobased hydrogels combined for metal ions removal. Ning et al. proposed hydroxyethyl cellulose hydrogel as a carrier material for the tannic acid to improve the adsorption performance and immobilize the tannin [46]. Pei et al. simultaneously achieved both tannin immobilization and hydrogel formation [47]. Nevertheless, these tannin-immobilized hydrogels are not prepared following the green chemistry principles using non-toxic substances. To date, there has been no detailed investigation of the decontamination of heavy metals from water using tannin supported on photocrosslinked natural hydrogels (Table 1).

In the current work, we exploited the green method of UV-curing to prepare the methacrylated chitosan-based hydrogels and the tannic acid capacity to interact with chitosan via hydrogen bonding [48,49], swelling the hydrogels in the tannin solution. Tannic acid contains five di-galloyl ester groups bonded to a glucose core [50]. Due to its unique structural features, it is capable of complexing or crosslinking macromolecules, such as chitosan, via multiple interactions, including hydrogen and ionic bonding and hydrophobic interactions [51]. Moreover, the galloyl groups prove binding sites for the coordinated interaction with several metal ions [50,52]. Additionally, it is a simple molecule that can be used as a model for several polyphenols (e.g., natural hydrolysable tannins which can be of interest for a green and circular approach in future works).

We compared the two types of hydrogels using FTIR spectroscopy, SEM, and Folin–Ciocalteu analyses. Moreover, the mechanical properties were assessed through compression tests before and after the tannic acid introduction into the chitosan-based hydrogel. Mechanical resistance is a parameter that influences the number of times the adsorbent material can be reused. Finally, another purpose of this investigation was to deeply explore the adsorption efficiencies of the two kinds of hydrogels towards copper ions, as an example of pollutant heavy metal, as a function of the pH, as well as the hydrogel’s reusability, a key factor for economizing the decontamination according to the requirements of a circular economy.

**Table 1 polymers-14-04645-t001:** Application of hydrogels for wastewater treatment.

Hydrogel	Active Agent	Ion/Ions Adsorbed	Solution and pH Range	Mechanical Properties	Reusability	Reference
Chitosan-based hydrogel beads	Chitosan	Cu(II)	0.157 mmol_Cu_/dm^3^ and 15.7 mmol_Cu_/dm^3^ CuSO_4_ pH = 3.5 and 5	-	-	[32]
Chitosan/PVA hydrogel beads	Chitosan	Pb(II)	30 mg/L PbCl_2_ pH range 2–8	-	2 sets of adsorption experiments	[53]
Chitosan hydrogel beads	Chitosan	Pb(II) Experiments regarding co-adsorption of Pb(II) and humic acid show a decrease in adsorption of Pb(II)	15 mg/L Pb(NO_3_)_2_ pH = 5.0, 6.5 and 7.5	-	-	[54]
Chitosan and aminated chitosan beads	Chitosan	Hg(II)	2–100 ppm Hg(NO_3_)_2_ pH range 3–9	-	-	[55]
CR-impregnated chitosan hydrogel	Chitosan	Cu(II)	317.5 mg/L C_4_H_8_CuO_5_ pH 2.0–6.1	-	3 cycles of adsorption-desorption	[56]
UV-cured chitosan-based hydrogel	Chitosan	As(V), Pb(II)	10–20 mg/L Na_2_HAsO_4_·7H_2_O, 50–75 mg/L Pb(NO_3_)_2_ pH range 2–9	-	-	[57]
Collagen/cellulose hydrogel beads	Collagen	Cu(II)	4.5 mmol/L CuSO_4_·6H_2_O pH range 1–6	-	4 cycles of adsorption-desorption	[58]
Tannic acid-based hydrogel	Tannic acid	Cu(II)	CuSO_4_·5H_2_O	-	-	[59]
Collagen tannin resin		Cu(II)	1 mmol/L CuSO_4_·6H_2_O pH range 2–5.5	-	-	[60]
Hydroxyethyl cellulose hydrogel modified with tannic acid	Tannic acid	Methylene blue	200–1200 mg/L methylene blue pH range 2–12	-	5 cycles of adsorption-desorption	[46]
PVA/alginate-based hydrogel with green tea waste	Polyphenols	Cu(II), Cr(VI)	2–100 mg/L CuSO_4_, 2–200 mg/mL K_2_Cr_2_O_7_ pH range 2–6	-	-	[61]
Tannin-immobilized cellulose hydrogel	Tannins	Methylene blue	10–80 mg/L methylene blue pH range 2–8	-	6 recycling events	[47]
Chitin-based composite hydrogel reinforced by tannic acid functionalized graphene (TRGO)	Tannic acid	Congo red	100–400 mg/L Congo red pH range 4.5–9.0	Compressive strength: 22.7 kPa (without TRGO) 72.3 kPa (with 7% of TRGO)	-	[62]

## 2. Materials and Methods

### 2.1. Materials and Chemicals

Low molecular weight chitosan (CH) (Mw = 50 kDa, ≥75% deacetylation degree), methacrylic anhydride (MA, 94%), acetic acid (96%), and Irgacure 2959 were purchased from Sigma-Aldrich (Milano, Italy). Tannic acid (TA), copper (II) sulphate [CuSO_4_], sodium carbonate [Na_2_CO_3_], and Folin–Ciocalteu phenol reagent 2 M (with respect to acid, 47,641) (F&C reagent) were also purchased from Sigma-Aldrich (Milano, Italy). To detect the copper ion concentration, Copper LR reagents composed of bicinchoninic acid disodium salt (HI9547-01, Hanna Instruments, Villafranca Padovana, Italy) were used. All of the materials were used without additional purification, and all solutions were prepared in ultrapure water.

### 2.2. Preparation of Hydrogels

#### 2.2.1. Synthesis of Methacrylated Chitosan (MCH)

The chitosan (CH) methacrylation was performed as previously described [63,64]. A total of 1.5 wt% of CH was dissolved in 2 wt% acetic acid solution and stirred. Once the solution was homogenous, MA was added dropwise according to the ratio between aminoglucose moieties and MA equal to 1:20. The reaction was carried out for 4 h at 50 °C under stirring conditions. The solution was dialyzed against distilled water for 4 days using a dialysis tubing cellulose membrane (MWCO = 14 kDa) and then freeze-dried to obtain MCH.

#### 2.2.2. UV-Curing of Chitosan-Based Hydrogel (CHg)

The freeze-dried MCH (1.5 wt%) was solubilized in 2 wt% acetic acid solution for 12 h. Then, 1 phr of Irgacure 2959 was added to the solution as a photoinitiator, and the mixture was stirred until homogeneity was achieved. The liquid formulation was poured into a silicon cylinder mold to obtain cylinder-shaped hydrogels (1 cm in diameter, 1 cm in the height), and the mixture was irradiated for 5 min with UV light (100 mW/cm^2^). A Hamamatsu LC8 lamp with an 8 mm light guide and spectral distribution range of 240–400 nm was used [38,57]. Finally, the produced CHgs were air-dried.

#### 2.2.3. Preparation of CHg Containing TA (CHg-TA)

Air-dried CHgs were dipped in a tannic acid (TA) solution with a concentration of 15 mg/mL and swelled for 3 h. The swelling in the TA solution was performed at a constant temperature of 37 °C. Then, the obtained hydrogels containing TA (CHg-TA) were washed three times with ultrapure water and air-dried.

### 2.3. Fourier Transform Infrared (FTIR) Spectroscopy

In order to confirm the successful methacrylation of CH and TA adsorption into CHg, attenuated total reflectance-infrared spectroscopy (ATR-FTIR) was used. The experiments were performed on air-dried hydrogels using a Thermo Scientific Nicolet iS50 FTIR spectrometer (Thermo Fisher Scientific, Milano, Italy) equipped with a diamond crystal ATR accessory. ATR spectra were collected with a resolution of 4 cm^−1^ in the range of 4000–600 cm^−1^.

### 2.4. Surface Characterization with SEM

The surface morphology and semi-quantitative chemical composition of air-dried hydrogels, before and after treatment in TA solution, were investigated by scanning electron microscopy (SEM, JEOL, JCM-6000 plus). The accelerating voltage was set at 15.0 kV.

### 2.5. Folin–Ciocalteu Assay

The total phenolic content in the hydrogels was measured using the Folin–Ciocalteu assay [65,66]. The TA detection in the air-dried hydrogels and solutions was performed by monitoring the reaction between the TA in the sample and phosphotungstic/phosphomolybdic acid contained in the F&C reagent. The test is based on the reduction of the F&C reagent caused by the oxidation of the phenolic compounds [67].

To detect TA in the air-dried hydrogels, CHg and CHg-TA were dipped in 8 mL of double-distilled water, then 0.5 mL of F&C reagent and 1.5 mL of 20 *w*/*v*% Na_2_CO_3_ were added. The reaction was conducted for 2 h in the dark. As far as solutions are concerned, 2 mL of the sample were poured into 6 mL of double-distilled water and then mixed with 0.5 mL of F&C reagent and 1.5 mL of 20 *w*/*v*% Na_2_CO_3_ and allowed to react for 2 h in the dark.

The photometric tests were performed by measuring the absorbance of the resulting blue color solution at 760 nm with a UV-Vis spectrophotometer (UV-2600, Shimadzu, Japan). The concentration of total phenols was quantified in accordance with a standard calibration curve of gallic acid, obtained by seven solutions with defined concentrations (0.001, 0.003, 0.005, 0.010, 0.020, 0.030, and 0.040 mg/mL), and the phenols are quantified by using gallic equivalent units (GAE) as a measuring unit.

### 2.6. Compression Test

Unconfined uniaxial compression tests were performed with an MTS QTestTM/10 Elite controller using TestWorks^®^ 4 software (MTS Systems Corporation, Edan Prairie, MN, USA). The cylindrical geometry of each sample was measured before testing (φ = 11 mm, h = 12 mm for CHg; φ = 4 mm, h = 4 mm for CHg-TA). The hydrogel was compressed at a test speed of 0.5 mm/min with a cell of 10 N. The data acquisition rate was set at 20 Hz. The compressive modulus (E) was estimated as the slope of the linear region of the stress–strain curves. All experiments were performed in triplicate.

### 2.7. Zeta Potential

The zeta (ζ) potential on the surface of CHg and CHg-TA was measured using an electrokinetic analyzer (SurPASS, Anton Paar) equipped with an automatic titration unit. The hydrogels were swelled in distilled water overnight, and about 500 mg of the swelled samples were inserted into a cylindrical cell. The ζ potential was assessed as a function of pH in an electrolyte solution of 0.001 M KCl. The pH was changed by the addition of 0.05 M NaOH and 0.05 M HCl solutions for titration in the basic and acid range, respectively. Fifteen points were measured for each range, and four ramps were performed for each point. The same sample was used for basic and acid titrations, but it was thoroughly rinsed with ultrapure water between the two measurements.

### 2.8. Adsorption Experiments

The Cu^2+^ solution was prepared from copper sulphate CuSO_4_ with a concentration of 10 ppm. To assess the effect of pH on Cu^2+^ ion adsorption, four different solutions were prepared, with pH equal to 2, 4, 6, and 10. The pH was adjusted with 0.1 M HCl or 0.1 M NaOH.

The air-dried CHg-TAs underwent a pre-washing treatment in distilled water for 4 h under stirring, according to a ratio of 7 mg:10 mL, before starting the copper ion adsorption experiments. The CHg-TA pre-washing step is useful to remove excess TA weakly bound to chitosan. The air-dried CHg and pre-washed CHg-TA were dipped in the metal ion solution with a ratio of 7 mg:10 mL. The adsorption experiments were conducted for 24 h at room temperature under mild conditions (300 rpm). Each experiment was conducted in triplicate, and the mean values were reported.

The total Cu^2+^ concentration in the solution was monitored using a colorimetric method based on the use of bicinchoninic acid disodium salt. The complex absorbance was detected using a Copper Low-Range Portable Photometer (HI96747, resolution of 0.001 mg/L, Hanna Instruments) equipped with a tungsten lamp as a light source and a silicon photocell with a narrow band interference filter at 560 nm as a detector. The Cu^2+^ determination was conducted at a pH range between 6.0 and 8.0. The adsorbed amount was measured by the difference between the initial and final concentrations. The adsorbent capacity (Q_e_ [mg/g]) and the removal efficiency (%R) were calculated according to the following equations.
(1)$$Q_{e}=\frac{\left(C_{i}-C_{f}\right)\times V}{W}$$
(2)$$\%R=\left(\frac{C_{i}-C_{f}}{C_{i}}\right)\times 100$$

C_i_ (mg/L) is the initial copper ion concentration, while C_f_ (mg/L) is the final metal ion concentration after 24 h of contact time. V (mL) is the volume of the Cu^2+^ solution, and W (g) is the mass of the air-dried hydrogel.

### 2.9. Recycling Experiments

The reusability and re-adsorption ability of hydrogels were tested to assess their stability as adsorbents. The desorption step was carried out by immersing the air-dried copper-loaded hydrogels in a 0.1 M HCl solution with a ratio of 7 mg:10 mL. The removal of adsorbed copper ions from the hydrogels was conducted for 90 min at room temperature under stirring. After elution, the desorbed hydrogel was washed with distilled water to remove any trace of acidity. The procedure of the adsorption-desorption cycle was repeated five times in the pH range of 2–6.

## 3. Results and Discussion

### 3.1. Characterization of CHg and CHg-TA

Figure 2a reports the FTIR spectra of raw CH and MCH. The successful methacrylation reaction was confirmed by the presence of a peak at 1720 cm^−1^, assigned to the C=O stretches. Furthermore, the signals at 1615 cm^−1^ and 800 cm^−1^ are absent in the CH spectrum and can be attributed to the C=C stretching vibration and C=CH_2_ bending vibration, respectively. After the incorporation of photocrosslinkable groups, the peak assigned to the N-H bending vibration shifted from 1582 cm^−1^ to 1538 cm^−1^. Thus, this red shift suggests an N-methacrylation [63]. In addition, the slight decrease in the characteristic wide band of the -OH vibrations around 3300 cm^−1^ proposes that the reaction involving the hydroxyl groups can occur.

In Figure 2b the spectra of CHg, TA powder, and CHg-TA are compared, proving the interactions between tannic acid and chitosan-based hydrogel. The band around 3300 cm^−1^ in the CHg spectrum is characteristic of the bond stretching of O-H and N-H. This region is stronger and wider in the TA spectrum and is assigned to the polyphenolic groups and aromatic C=C. Therefore, the intensity of this band in the CHg-TA spectrum broadened due to the formation of hydrogen bonds between TA and CHg. After CHg was swelled in the TA solution (Figure 3a), the shift of the C=O peak in the CHg spectrum to a lower wavenumber was also detected at 1708 cm^−1^, suggesting the presence of the hydrogen-bonding interaction. This frequency is also assigned to the carbonyl groups of the TA. The peak at 1178 cm^−1^ in the TA spectrum can be attributed to the C-O bonds of the ester group [68], and its red shift at 1153 cm^−1^ confirms the TA presence within CHg. The red shift may be explained by the formation of hydrogen bonds involving the hydroxyl and carbonyl groups, suggesting that TA was successfully immobilized within CHg. Moreover, the spectral signals at 1608 cm^−1^ and 1445 cm^−1^ are absent in the CHg spectrum and are characteristic of stretches of the aromatic C=C bonds of the TA. The strong band at 1050-1030 cm^−1^ can be assigned to stretching vibrations of the pyranose ring C-O bond in CHg [69] and the ether C-O bond in TA. This region broadened in the CHg-TA spectrum exhibiting a higher intensity. Therefore, the ATR-FTIR analyses confirm the formation of strong interactions between the pyrogallol and catechol groups in TA and the functional groups of the polysaccharide, reinforcing CHg [70,71] (Figure 3b).

After the CHg swelling in the TA solution, the color change of the hydrogels from light yellowish to brownish (Figure 3c) suggests the formation of interactions between TA and CHg. The F&C results prove the successful adsorption of TA by CHg, and the total phenolic content in CHg-TA was measured at 0.094 ± 0.07 mg/mL. Therefore, no redox activity of CHg was detected, as expected by the chitosan structure.

Moreover, the surface morphology of the produced hydrogels was observed by using SEM (Figure 4). It was found that the surfaces of both CHg and CHg-TA are irregular, and the TA introduction does not cause any morphological changes to the hydrogel. The acquired SEM image of CHg (Figure 4a) shows a less rough surface than that of the CHg-TA (Figure 4b). The rougher surface of CHg-TA determines a higher surface area, a parameter that may influence the adsorption efficiency. The pores were not observed on the samples surfaces.

Lastly, the hydrogels were subjected to unconfined uniaxial compression. The compression tests revealed that the introduction of TA in hydrogels substantially improves their mechanical performance, as He et al. confirmed in their previous work [70]. Specifically, the resulting compressive modulus (E) is 7.2 ± 1.8 kPa and 0.94 ± 0.1 MPa for CHg and CHg-TA, respectively. This exceptionally substantial increase in CHg-TA stiffness confirms the higher crosslinking density due to a high number of formed hydrogen bonds, resulting in an important reinforcing effect of the UV-cured hydrogel.

### 3.2. Copper Ions Adsorption from Water

The UV-cured chitosan-based hydrogels were tested for copper ion removal from water. The effect of pH variation from 2 to 10 toward CHg and CHg-TA adsorption capacity was investigated. The pH is a fundamental parameter to be controlled, as the ionic state of the surface functional groups of hydrogels is a function of pH, and this play an important role in hydrogel and metal ion interaction.

Chitosan exhibits a pKa value close to 6.5 due to its primary amino groups [73]. Hence, at pH above 6.5, the amino groups are in the form of -NH_2_, improving the attraction between polymeric chains due to van der Waals interactions. However, at a pH below pKa, the amino groups are protonated, producing an increment in positive charges on the CHg surface, thus causing an electrostatic repulsion among the chitosan polymeric chains. The ζ potential measurement of the CHg surface charge (Figure 5a) confirms this trend, identifying the isoelectric point (IEP) at a pH of about 6.3, an index of the prevalence of basic groups on the surface. The presence of basic functional groups is confirmed by the plateau between 4 and 5. As a result, at acid conditions, the positive surface reduces the diffusion of metal cations into CHg, leading to a decrease in the interactions between Cu^2+^ ions and chitosan [74]. As revealed by FTIR-ATR, chitosan also possesses acidic functional groups (-OH groups) evidenced by the plateau in the basic region of the zeta potential curve.

In order to enhance the CHg adsorption capacity, the now-known tannic acid ability to chelate metal ions is exploited [43,59,75], reinforcing chitosan-based hydrogel with TA introduction through the multiple hydrogen-bonding interactions. By inserting the polyphenols into the CHg structure, the number of -OH groups increases, thus raising the number of available sites for coordinating the Cu^2+^ ions. The TA introduction slightly reduces the IEP of CHg-TA at a pH of 6.0 (Figure 5a) due to the introduction of a significant number of acidic functional groups (-OH). The presence of acidic OH is confirmed by the more pronounced plateau in the basic region, compared to the curve of CHg. The plateau in the acidic region reveals the permanence of the chitosan basic functional groups, even after TA functionalization.

To minimize the TA release into the Cu^2+^ ion solution and to avoid further pollution, the CHg-TA was swelled in distilled water for 4 h (pre-washing). The maximum absorption wavelength at about 280 nm in the UV-Vis spectrum (Figure 5b) is a noticeable evidence of TA presence in the water used for swelling. The comparison between the UV-Vis spectra of CHg and CHg-TA after swelling in water proves that the tannin release is not quantified by the Lambert–Beer law due to the low contribution of the photoinitiator. Indeed, the maximum absorbance peak derived from Irgacure 2959 is determined to be in the same range as TA absorbance. The tannin released into the pre-washing solution can be measured by the F&C assay, and the CHg-TA released 0.003 ± 0.0006 mg/mL of phenols (measured as GAE).

Figure 5c illustrates the Cu^2+^ adsorption of the CHg and CHg-TA as a function of pH. The experimental data provide evidence that the tannic acid in chitosan-based hydrogel enhances the adsorption capacity at acid conditions. Since the chitosan surface shows a positive ζ potential, mainly due to the protonated amino groups, repulsive electrostatic interactions are generated between CHg and the positively charged metal ions. Therefore, in the acid pH range, the amino groups are not available sites for the Cu^2+^ adsorption and may hinder Cu^2+^ cations from approaching the CHg surface. Thanks to the greater number of -OH groups introduced with TA, the Q_e_ values increase due to the coordination with Cu^2+^ ions. Specifically, the two adjacent phenolic hydroxyls (catechol) can coordinate the metal ions (Figure 5d) [77,78,79]. The greatest difference in the adsorption capacity between CHg and CHg-TA can be noticed at pH 2, and the Q_e_ is approximately 11.04 ± 0.48 mg/g for CHg-TA, while it is approximately 9.96 ± 1.22 mg/g for CHg.

During the decontamination treatment, the tannin release caused by CHg-TA can occur due to the high solubility of TA in water. The amount of TA in the copper solutions was quantified after the adsorption process, and the results of F&C testing for the copper solutions at a pH of 2, 4, and 6 are reported in Figure 6. It can be seen from these data that at pH 4 and 6, the TA release is almost negligible, and this release is the result of the pre-washing step of CHg-TA. Therefore, at these two pH values, the CHg-TA can improve the copper adsorption without significant further contamination of the water by the tannin release. Under extreme acidic conditions, such as at pH 2, TA can be partially or totally hydrolyzed, reducing the hydrogen bonds with chitosan and releasing gallic acid [40,80,81].

Even if the CHg-TA surface shows a negative ζ potential as alkalinity raises, the CHg-TA Q_e_ slightly decreases, possibly due to the oxidation of the polyphenol groups to the quinone form over time [82,83]. Once quinones formed, many hydrogen bonds were destroyed. As a result, the network between CHg and TA decreases [48], and tannin release can occur in the Cu^2+^ solution.

In contrast to the adsorption capacity of CHg-TA, CHg shows the highest Q_e_ towards Cu^2+^ cations at a basic pH, as its surface possesses a negative ζ potential [57,74]. This finding suggests that the deprotonation of -OH groups allows for the adsorption of the Cu^2+^ ions through electrostatic interactions, while the neutral amino groups chelate the metal ions.

### 3.3. Regeneration Capability and Reusability

The adsorbent regeneration capability is one of the important properties of practical wastewater treatment [84]. The adsorption stability of CHg and CHg-TA was studied in the pH range of 2–6, where CHg-TA shows higher Q_e_ than CHg. The regenerated hydrogels were used to adsorb the same concentration of CuSO_4_ (10 ppm). As shown in Figure 7, the CHg-TA removal efficiency (% R) was just slightly reduced after 5 adsorption-desorption cycles. The % R decrease may be attributed to the incomplete desorption of Cu^2+^ from the hydrogel and the progressive tannin release into the treated solutions. However, the Cu^2+^ removal of CHg-TA was still higher than 60% for all three pH levels studied. Rather, CHg is not reusable after 2 cycles due to the destruction of the polymeric network, mainly at extremely acidic pH levels, such as those in the desorption solution. This result can be assigned to the electrostatic repulsion among the polymeric chains formed by the increase in protonated amino groups.

The excellent stability and recycling ability of CHg-TA may be related to its mechanical properties. The greater stiffness introduced with TA allows for obtaining a more acid-resistant chitosan-based hydrogel. Therefore, the CHg-TA property of maintaining its structural integrity, thanks to its higher crosslinking density, increases its recyclability without remarkable % R losses.

## 4. Conclusions

This work aims to evaluate the possibility of successfully introducing tannic acid into a UV-cured hydrogel derived from a natural and renewable source to improve its mechanical properties, which can affect its adsorption capacity and allow for an increase in adsorbent reusability.

The copper ion adsorption capacities of CHg and CHg-TA as a function of pH were compared. Under acidic conditions, the chitosan’s amino groups are not available as adsorption sites, while the coordinating ability of the high number of polyphenols in CHg-TA contributes to enhance copper ion adsorption. 

From recycling experiments, CHg-TA was found to possess effective adsorption capacity and reusability at acidic pH, and it could be used for 5 cycles of adsorption-desorption, unlike CHg. The prominent stability of CHg-TA can be associated with the remarkably enhanced mechanical properties derived by the increased crosslinking density due to the formation of hydrogen bonds between the functional groups of chitosan and TA. The outstanding regeneration capability of CHg-TA may reduce the cost of the decontamination treatment of acid industrial wastewater and make it more environmentally friendly. 

## Figures and Tables

**Figure 1 polymers-14-04645-f001:**
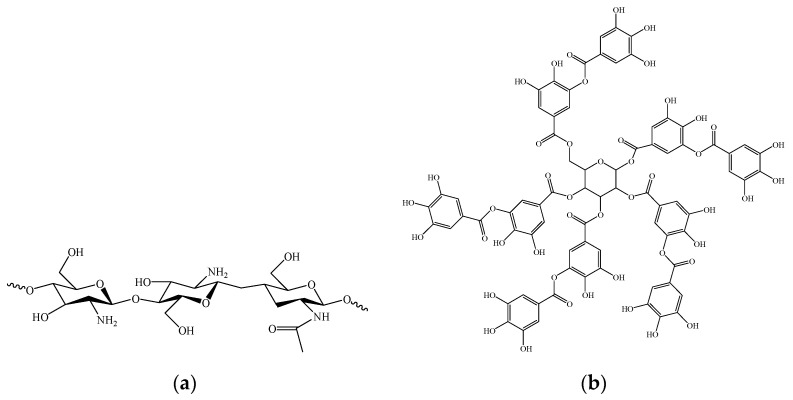
Molecular structures of (**a**) chitosan and (**b**) tannic acid.

**Figure 2 polymers-14-04645-f002:**
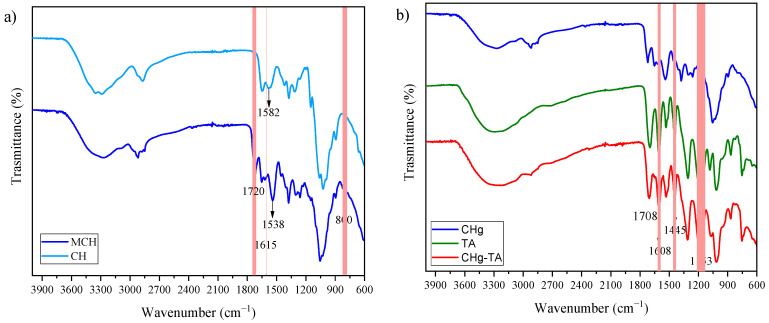
FTIR spectra of (**a**) chitosan before and after the methacrylation reaction and (**b**) CHg-TA, compared to CHg and TA powder.

**Figure 3 polymers-14-04645-f003:**
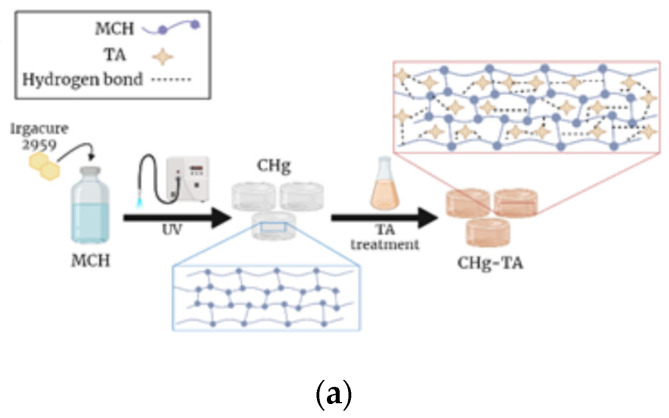
(**a**) Representative scheme of the CHg and CHg-TA preparation; (**b**) formation of hydrogen bonds between MCH functional groups and TA; (**c**) digital photo of CHg and CHg-TA, showing that the TA adsorption can be visually confirmed by the formation of the typical light brown color of TA [72].

**Figure 4 polymers-14-04645-f004:**
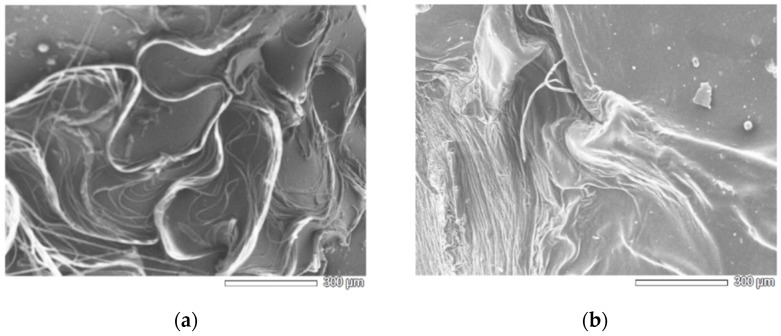
SEM images on a representative macro area (100× *g* mag.) of (**a**) CHg and (**b**) CHg-TA.

**Figure 5 polymers-14-04645-f005:**
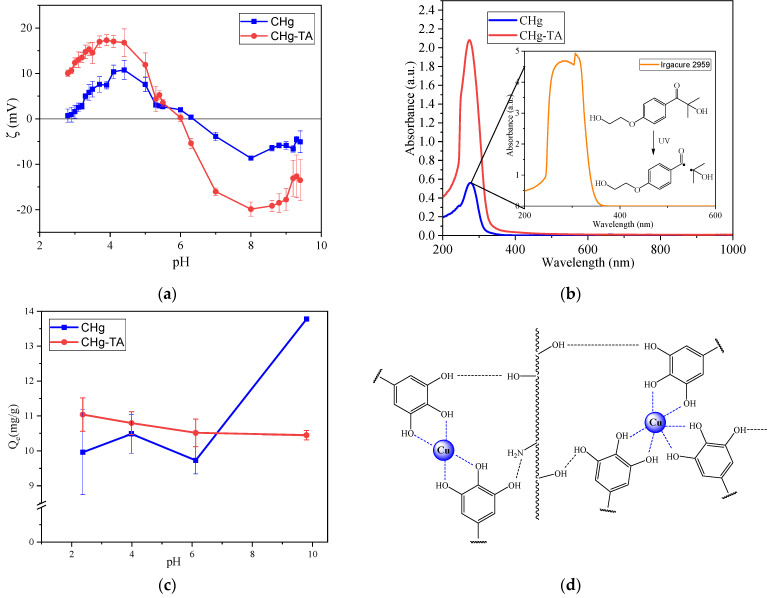
(**a**) ζ potential curves of CHg and CHg-TA; (**b**) UV-Vis spectra of pre-washing solutions and Irgacure 2959, with the mechanism of the benzoyl and alkyl radicals formation [76]; (**c**) the influence of pH on the Q_e_ values of Cu^2+^; (**d**) schematic representation of the adsorption mechanism of Cu^2+^ into CHg-TA.

**Figure 6 polymers-14-04645-f006:**
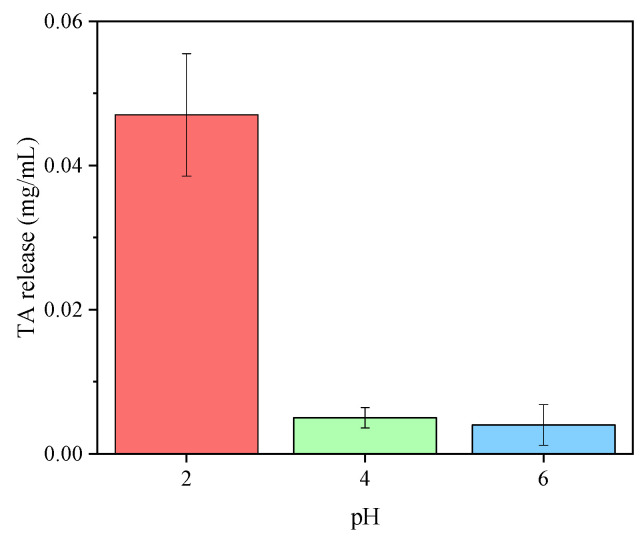
Folin–Ciocalteu determination of TA in the CuSO_4_ solutions after 24 h of adsorption treatment by CHg-TA.

**Figure 7 polymers-14-04645-f007:**
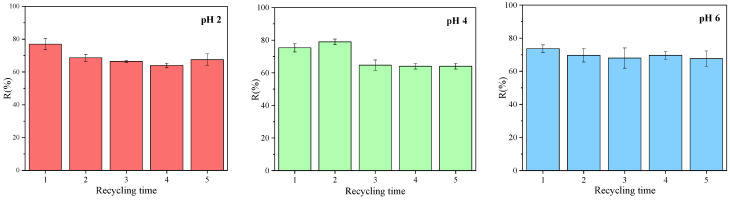
Re-adsorption efficiency of CHg-TA at different pH levels.

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
