# Peer review of "UV-Cured Chitosan-Based Hydrogels Strengthened by Tannic Acid for the Removal of Copper Ions from Water"

_polymers, 2022, doi:10.3390/polym14214645_

Round 1

Reviewer 1 Report

The paper "UV-Cured Chitosan-Based Hydrogels Strengthened by Tannic Acid for Removal of Copper Ions from Water" is concentrated on the preparation of a new material with adsorbing properties. The material based on chitosan and tannic acid was easily obtained and its characteristics are very promising. Authors have proved along their research that this material is able to successfully retain copper ions from water. Moreover, the adsorbent can be used multiple times.

The paper is of great interest. It is well written, and it discusses all the significant aspects implied. 

Only some minor corrections in terms of text editing (e.g., page 5, line 168: the subscript is missing on sodium carbonate formula; page 5, lines 199-200: "The pH was adjusted with 0.1 M HCl or (not and) 0.1 M NaOH" etc.) are required.

Author Response

we have upload the comments

Reviewer 2 Report

In this work UV-cured chitosan-based hydrogels strengthened by tannic acid for the removal of copper ions from water are described. The hydrogel preparation was performed via a simple two-step method of photocrosslinking of methacrylated chitosan and subsequent swelling in the TA solution. In addition hydrogels reusability was assessed. The work is of interest because according to obtained results CHg-TA is as a promising green candidate for heavy metal ions removal from mainly acid wastewater. The article looks like a short communication and may be published after major revision.

Notes:

1. Why authors did chose tannic acid among hydrolysable tannins for this research? It should be reflected in the Introduction.

2. Usually gel permeation chromatography is applied for characterization of polymers. Why authors did not use this method for characterization of a chitosan-based hydrogel containing tannic acid (CHg-TA) obtained in this work?

3. Did authors investigate the developed chitosan-based hydrogel for the removal of other heavy metal ions? How will it behave in a mixture of several heavy metals?

4. Conclusions should be shortened and focused on the main findings of this work. Promising application of new obtained results also should be added.

Author Response

We have upload the comments

Reviewer 3 Report

The article titled “UV-Cured Chitosan-Based Hydrogels Strengthened by Tannic Acid for the Removal of Copper Ions from Water” by Sesia et al. has been reviewed where they have designed material for heavy metal removal from the acid wastewater and reported enhanced adsorption efficiency of the chitosan-based hydrogel which is photocurable. The materials used are renewable and from the natural sources. The works has importance in the area and can bring new insights, however a few of the concerns are to be addressed prior to the work is considered.

1.       Line No. 136: What is the MWCO value of the dialysis tube used?

2.       Line No: 263 & Figure 3c: Possible reason to describe the change in color from light yellow to brown may be provided.

3.       The selectivity of Cu-ion removal in presence of other heavy metal ions could be provided to show its efficiency to remove Cu ions even when other ions are present in the acidic wastewater.

4.       There are plenty of studies reported so far on selective separation of Cu ions from wastewater, how is the designed material efficient in comparison are to be provided in a table.

5.       “Author Contribution” section needs correction.

In my opinion, the novelty of this work has to be improved in comparison to the other materials already reported.

Author Response

We have upload the comments 

Round 2

Reviewer 2 Report

In this work UV-cured chitosan-based hydrogels strengthened by tannic acid for the removal of copper ions from water was developed. The hydrogel preparation was performed via a simple two-step method of photocrosslinking of methacrylated chitosan and subsequent swelling in the TA solution. The obtained hydrogels were characterized using ATR-FTIR, SEM, and Folin&Ciocalteu assay. The work is of interest because the developed chitosan-based hydrogel containing tannic acid can be applied for multiple cycles of adsorption and desorption during the decontamination of acid industrial wastewater.

Authors have corrected all comments in the paper and quite clearly answered to the questions. I hope these corrections improved the paper and the revised version corresponds to high standards of Polymers. After careful consideration, I think that this article may be published in this view. 

Author Response

The answer is reported in the attached file

Reviewer 3 Report

The authors have tactfully declined to improve the work by not following the two main suggestions of the reviewer which could have improved the novelty of this work.

In response to the comment # (3), the authors have mentioned that the suggestion of providing the selectivity study of their designed material will be a future work, whereas it was highly recommended to investigate the suitability of this tannic acid introduced hydrogel for the competitive adsorption of Cu ions. There are plenty of published works that have used polymers and their composite materials from natural sources for heavy metal removal from wastewater, hence just the introduction of tannic acid into the polymer gel and using it for Cu removal does not present this work with novelty. Unfortunately, I do not fine something very much new in this work. The comment of the reviewer had given them an opportunity to improve the work, but the authors have opted to not to carry out it, whereas any good paper would show the suitability of such materials on how efficient their material is to carry out the competitive adsorption. Thus, it is not clear from this work that their material has sufficient novelty to warrant publication in this mainline journal of Polymers which has almost all the polymer researchers as its readers.

The suggestion in comment # (4) was also an opportunity to show that this so designed material is superior to many/few other materials designed by other researchers. The reviewer is well aware that this is not a review work, but the authors have not shown whether their material is a superior one. Since, the authors do not want to compare the efficiency of their materials with the efficiency of other good materials designed for Cu-ion removal, I highly doubt the quality of this material and its possibility of use for practical purposes. Only the idea of incorporating tannic acid into chitosan gel is not sufficiently novel to attract warrant publication in Polymers.

In addition, in response to the comment # (2) on color change, the authors could have carried out some spectroscopic investigation (in addition to the digital image) to demonstrate the change and the based on the location of the peaks of the spectra they should have provided solid arguments to support the change and its reason. Providing only the digital image and trying to explain it is not a very scientific way of investigation, here.

Some other points:

In the response to reviewer, in addition to highlighting, authors should provide the line numbers and page numbers to notify the changes they have made in the revised version.

Authors should highlight all the changes they have made to attract attention of the reviewers. For example, authors have included the MWCO value in the revised manuscript, but neither highlighted nor referred by line number of page number, and the reviewer had to take extra effort search for it. It is to mention that the reviewers invest their time and effort voluntarily to contribute to improve the works and might have limited time to review a work.

The Table caption of the table is below the table instead of it keeping at the top of the table. Seems, authors need to pay more attention to prepare the manuscript.

Although, suggestions were given to improve the work, the authors have not done those accept including the other two minor comments. Only introducing tannic acid to chitosan and showing its use for Cu-ion removal does not show high novelty. Hence, I am sorry as I cannot recommend this article for publication in this mainline journal.

Author Response

(The authors gave the same response as above.)
